# Effect of Back Pressure on Performances and Key Geometries of the Second Stage in a Highly Coupled Two-Stage Ejector

**DOI:** 10.3390/e24121847

**Published:** 2022-12-18

**Authors:** Jia Yan, Yuetong Shu, Chen Wang

**Affiliations:** 1School of Civil Engineering and Architecture, Southwest University of Science and Technology, Mianyang 621010, China; 2School of Control Science and Engineering, Shandong University, Jinan 250061, China; 3Department of Mechanical, Aerospace and Civil Engineering, University of Manchester, Manchester M13 9PL, UK

**Keywords:** back pressure, highly coupled, two-phase ejector, optimization, geometries

## Abstract

In this paper, for a highly coupled two-stage ejector-based cooling cycle, the optimization of primary nozzle length and angle of the second-stage ejector under varied primary nozzle diameters of the second stage was conducted first. Next, the evaluation for the influence of variable back pressure on ER of the two-stage ejector was performed. Last, the identification of the effect of the variable back pressure on the key geometries of the two-stage ejector was carried out. The results revealed that: (1) with the increase of the nozzle diameter at the second stage, the ER of both stages decreased with the increases of the length and angle of the converging section of the second-stage primary nozzle; (2) the pressure lift ratio range of the second-stage ejector in the critical mode gradually increased with the increase of the nozzle diameter of the second-stage; (3) when the pressure lift ratio increased from 102% to 106%, the peak ER of the second-stage decreased, and the influence of the area ratio and nozzle exit position of the second-stage ejector on its ER was reduced; (4) with the increase of nozzle diameter of the second-stage, the influence of area ratio and nozzle exit position of the second-stage on the second-stage performance decreased; and (5) the optimal AR of the second stage decreased but the optimal nozzle exit position of the second stage kept constant with the pressure lift ratio of the two-stage ejector.

## 1. Introduction

Nowadays, we are facing global warming and resource shortage [1]. Therefore, energy-saving and environmentally friendly refrigeration technologies have become a topic of widespread interest to refrigeration researchers and practitioners [2]. Ejector-based refrigeration systems have some advantages, such as no moving parts, waste heat driving and low operating costs [3,4]. Undoubtedly, ejector-based refrigeration systems are a promising industry [5,6].

In recent years, scholars have engaged in the research and development of two-stage ejector-based refrigeration systems. Kong et al. [7] investigated a supersonic two-stage ejector-diffuser system with numerical methods, and the system had four times better performance than a single ejector system. In addition, Kong et al. [8] predicted the flow phenomenon inside the two-stage ejector-diffuser. The results showed that the entrainment effects of the system greatly increased. Liu et al. [9] proposed a modified transcritical CO_2_ ejector enhanced two-stage compression cycle. The results showed that the heating coefficient of performance (COP) of the cycle, and so on, outperformed others. Wang et al. [10] presented a gas-fired air-to-water ejector heat pump. The system performance was improved with a high entrainment ratio. Liu et al. [11] proposed a novel two-stage compression transcritical CO_2_ refrigeration system with an ejector. The results indicated that the performance of the novel system was better than those of conventional systems. Yan et al. [12] presented a dual-ejector refrigeration system, and the area ratio (AR) had the most significant influences on the performance of the two-stage ejector. Ierin et al. [13] optimized a hybrid two-stage CO_2_ ejector-based cooling system, and the efficiency of the system increased by up to 32.7%. Ghorbani et al. [14] investigated a two-stage ejector cooling system, and the consumed power of the system decreased by 12.37%. Cao et al. [15] proposed a two-stage evaporation cycle and their numerical results disclosed that the COP of the cycle was improved, and the exergy was reduced. Sun et al. [16] claimed that the influence of phase transition in the ejector contributed to the ejector optimization. Chen et al. [17] studied the effect of the second-stage geometrical factors on the system performance and the optimized length to diameter ratio was 5. Similarly, Yadva et al. [18] numerically analyzed the performance of a two-stage ejector. Yang et al. [19] evaluated the exergy destruction characteristics inside a transcritical CO_2_ two-stage refrigeration system and showed that the system exergetic performance can be improved by enhancing the efficiency of the ejector. Yang et al. [20] also found that the gas cooler temperature had the greatest influence on ejector performance. Surendran et al. [21] explored a novel transcritical ejector regenerative refrigeration system and identified the system performance. Asfahan et al. [22] presented a system with two-stage ejectors and investigated them numerically. The results showed that the system had good performance when using the two-stage ejectors. Ding et al. [23] performed numerical studies using computational fluid dynamics (CFD) to predict two-stage ejector performance for subzero applications. Manjili et al. [24] used Engineering Equation Solver (EES) software to investigate a two-stage transcritical CO_2_ refrigeration cycle. It was found that the COP of new cycle was improved from 20% to 80% compared to the conventional cycle. Xue et al. [25] proposed and studied a two-stage vacuum ejector by comparing seven different ejector models. The results showed the two-stage ejector could provide superior suction pressure. Exposito-Carrillo et al. [26] optimized a two-stage CO_2_ refrigeration system, and the COP improved up to 13%. Wang et al. [27] developed a CFD model to investigate the performance of a proposed two-stage ejector. Viscito et al. [28] proposed a seasonal performance analysis of a hybrid ejector cooling system, they claimed that system required three or four ejectors for any reference climate, and they obtained an increase of the seasonal energy efficiency ratio up to 107%. Lillo et al. [29] presented a thermo-economic analysis of a waste heat recovery hybrid ejector cycle with a cooling load of 20 kW, the thermo-economic performance of this cycle has evident advantages over other waste heat driven system. Li et al. [30] carried out a numerical analysis of the influence of nozzle geometries on steam condensation and irreversibility in the ejector nozzle. The results indicated that the condensation of the steam makes a large amount of irreversible energy. Wen et al. [31] presented a two-stage ejector-based refrigeration system and optimized the two-stage ejector (TSE) geometries and system performance. Yan et al. [32] proposed another type of a highly coupled TSE-based system as shown in Figure 1, and they also optimized the key geometries such as area ratio (AR) and nozzle exit position (NXP) of the two-stage ejector, as illustrated in Figure 2, in which the mixture coming from the outlet of the first stage enters the primary nozzle of the second stage and entrains its secondary flow refrigerant.

However, no studies in the literature have mentioned the optimization of the key geometries, such as AR and NXP, of the highly coupled second-stage ejector under different back pressures and varied primary nozzles of the second stage (PNTD_2_). To bridge the gap, and based on our previous studies [33], further works in this study included:CFD modelling and model validation of the highly coupled TSE;Optimization of primary nozzle geometry of the second-stage ejector under varied primary nozzle of the second stage;Evaluation the influence of variable back pressure on entrainment ratio (ER) of the TSE;Identification of the effect of the variable back pressure on the key geometries of the second-stage ejector.

## 2. System Description and Numerical Method

### 2.1. System and Initial TSE Geometries

The schematic of the highly coupled TSE-based refrigeration system is shown in Figure 1, and the initial geometrical parameters of the TSE are presented in Figure 3.

### 2.2. CFD Modelling

The flow inside the TSE is calculated by using governing equations [33,34], and Gambit 2.4 and Ansys 19.0 [35] are used in this simulation. Grids with 103,000 quadrilateral elements are created as shown in Figure 4.

R134a is the working fluid with parameters from NIST [36], the RNG k-ε turbulence model was selected in this study. The standard wall function was chosen, and the range of the first grid cell is in the region of 30 < y+ < 300. The residual convergence limit for each equation is below 10^−5^, except that for energy equation is set to less than 10^−6^. In addition, to ensure that the refrigerant liquid completely evaporates into refrigerant gas, three inlet streams are set as 10 K superheat. The primary fluid inlet and the secondary fluid inlet are both set as the pressure inlet, while the outlet is set as the pressure outlet [32], and the boundary conditions of the TSE are illustrated in Table 1.

Three levels of grids (71,000, 103,000 and 138,000) are used to validate the grid independence as illustrated in Figure 5. Since the three grid levels are quite close with each other, the medium one is finally used in the following simulation.

## 3. Model Validation

### 3.1. Experimental Setup

The experimental setup is presented in Figure 6, in the setup, Evaporator 1 is simulated as an air conditioner, and its evaporating temperature is set as 7 °C; Evaporator 2 is simulated as a refrigerator, and its evaporating temperature is arranged as −5 °C; whist, Evaporator 3 is simulated as a freezer, and its evaporating temperature is specified as −30 °C. The ambient temperature is valued at 36 °C. Based on the thermodynamic calculation, the individual required cooling loads for three evaporators are 1566.2 W, 609.4 W and 997.2 W, respectively. Other details can refer to our previous study [32]. The range and accuracy of sensors are presented in Table 2.

### 3.2. Validation of the CFD Model

Fifteen CFD simulation results, as illustrated in Table 3, were validated by the experimental data. The average and maximum discrepancy of ER_1_ were 7.2% and 11.9%, and those for ER_2_ are 5.9% and 10.6%, respectively; therefore, the models can be used in the following simulations.

## 4. Results and Discussion

### 4.1. Optimization of Nozzle Geometry of the Second-Stage Ejector

#### 4.1.1. Optimization of Nozzle Length and Angle of the Second-Stage Ejector When PNTD_2_ Is 4.1 mm

Figure 7 shows the results of ER_1_ and ER_2_ with the second-stage ejector nozzle length (LC_2_) when PNTD_2_ is 4.1 mm. When LC_2_ changes from 5 mm to 40 mm, ER_1_ first rises from 0.595 to the maximum value of 0.646 (LC_2_ = 25 mm), and then slowly decreases to 0.636 (LC_2_ = 40 mm). ER_2_ rises slowly from 2.096 to 2.153 (LC_2_ = 20 mm) and then decreases rapidly until it reaches a minimum of 2.022. When LC_2_ changes from 5 mm to 40 mm, the maximum deviations of ER_1_ and ER_2_ reach 8.571% and 6.479%, respectively, which means that the change of LC_2_ has an impact on the performance of both the first stage and the second-stage, but the impact on the first-stage is more obvious. At the same time, it can be seen that when LC_2_ = 15–25 mm, the values of ER_1_ and ER_2_ are relatively large.

Figure 8 displays the results of ER_1_ and ER_2_ with the second-stage ejector nozzle angle (AC_2_) when PNTD_2_ is 4.1 mm. With the increase of AC_2_, ER_1_ first increases and then decreases, and the maximum value of ER_1_ is 0.653 (AC_2_ = 16°). As AC_2_ increases from 6° to 10°, ER_2_ increases from 2.146 to 2.171, and ER_2_ gradually decreases to 2.014 as AC_2_ continues to rise to 22°. Compared with the initial values of ER_1_ and ER_2_ (0.632 and 2.153), the maximum values of ER_1_ and ER_2_ increase by 0.021 and 0.019, respectively, and the maximum deviations of ER_1_ and ER_2_ are 11.054% and 7.795%, respectively. This means that AC_2_ has an impact on the performance of both stages, but ER_1_ is more sensitive to the changes of AC_2_. Moreover, it can be seen that AC_2_ has a greater impact on the two-stage performance than LC_2_. In addition, the AC_2_ range for which ER_1_ achieves large values is 10°–22°, while the AC_2_ range for which ER_2_ achieves large values is 6°–14°, indicating that the AC_2_ range for which both ER_1_ and ER_2_ obtain large values is 10°–14°.

#### 4.1.2. Optimization of Nozzle Length and Angle of Second-Stage Ejector When PNTD_2_ Is 4.7 mm

Figure 9 reveals the results of ER_1_ and ER_2_ with LC_2_ when PNTD_2_ is 4.7 mm. It can be seen that when LC_2_ changes from 5 mm to 45 mm, ER_1_ first rises from 1.031 to the maximum value of 1.098 (LC_2_ = 35 mm), and then rapidly decreases to 1.074 (LC_2_ = 45 mm). However, ER_2_ slowly increases to the maximum value of 1.861 (LC_2_ = 15 mm), follows by a rapid decline in ER_2_ until it decreases to the lowest value of 1.799. When LC_2_ changes from 5 mm to 45 mm, the maximum deviations of ER_1_ and ER_2_ reach 6.499% and 3.446%, respectively, which reflects that the performance of the first-stage and the second-stage are affected by the change of LC_2_, but the first-stage can be affected more obviously. Furthermore, it can be seen that when LC_2_ = 15–25 mm, the values of ER_1_ and ER_2_ are relatively large.

Figure 10 indicates the results of ER_1_ and ER_2_ with AC_2_ when PNTD_2_ is 4.7 mm. As shown in the figure, with the increase of AC_2_, ER_1_ increases first and then decreases, and the highest value of ER_1_ is 1.091 (AC_2_ = 14°). With the increase of AC_2_ from 6° to 8°, ER_2_ increases from 1.849 to 1.856; and with the increase of AC_2_ to 22°, ER_2_ decreases to 1.811. The maximum deviations of ER_1_ and ER_2_ are 5.106% and 2.485%, respectively, which means that AC_2_ has an impact on the performance of both stages, but ER_1_ is more sensitive to the change of AC_2_. In addition, the range of AC_2_ where both ER_1_ and ER_2_ are at large values is still within 10°–14°.

#### 4.1.3. Optimization of Nozzle Length and Angle of Second-Stage Ejector When PNTD_2_ Is 5.3 mm

Figure 11 presents the results of ER_1_ and ER_2_ with LC_2_ when PNTD_2_ is 5.3 mm. As illustrated in the figure, when LC_2_ changes from 5 mm to 40 mm, ER_1_ increases from 1.502 to the maximum value 1.585. ER_2_ decreases from 1.645 to a minimum of 1.598. Compared with the initial values of ER_1_ and ER_2_ (1.573 and 1.627), the maximum values of ER_1_ and ER_2_ increase by 0.012 and 0.018. In addition, when LC_2_ changes from 5 mm to 40 mm, the maximum deviations of ER_1_ and ER_2_ are 5.526% and 2.941%, respectively, which means that the performance of both stages is affected by the change of LC_2_, and the performance of the first-stage is slightly more affected. In contrast, when LC_2_ is 15–25 mm, the values of ER_1_ and ER_2_ are at large values.

Figure 12 shows the changes of ER_1_ and ER_2_ with AC_2_ when PNTD_2_ is 5.3 mm. It can be seen that with the increase of AC_2_, ER_1_ first increases and then decreases, and the maximum value of ER_1_ is 1.573 (AC_2_ = 14°). As AC_2_ increases from 6° to 10°, ER_2_ increases from 1.629 to 1.632, when AC_2_ continues to increase to 18°, ER_2_ gradually decreases to 1.621. The maximum deviations of ER_1_ and ER_2_ are 3.897% and 1.527%, respectively, which indicates that AC_2_ has an impact on the performance of both stages, but ER_1_ is slightly more sensitive to AC_2_. In addition, the range of AC_2_ is still within 10°–14° when both ER_1_ and ER_2_ are at large values.

In conclusion, both LC_2_ and AC_2_ have certain effects on the performance of the TSE. With the increase of PNTD_2_, the influence of LC_2_ and AC_2_ on the performance of the two stages is gradually weakened. With a comprehensive consideration, LC_2_ = 25 mm and AC_2_ = 12° are selected as the optimized geometries of the second-stage ejector converging nozzle to carry out the following study on the influence of variable back pressure on the ER of TSE.

### 4.2. Influence of Variable Back Pressure on ERs of the TSE

Boundary conditions of the TSE except the back pressure are kept constant, and the change of the back pressure of the TSE is expressed as the percentage of pressure lift, namely the change of PLR (the ratio of the back pressure to the secondary inlet pressure of the second-stage ejector). The initial PLR is 108%. When PNTD_2_ is 4.1 mm and PLR changes in the range of 102–118%, the influence of the changed PLR on ER_1_ and ER_2_ is shown in Figure 13. It can be seen that when PLR increases from 102% to 118%, ER_1_ gradually decreases, but its maximum value and minimum value are 0.644 and 0.643, respectively. Therefore, ER_1_ almost does not change. ER_2_ decreases almost linearly from 2.442 to 1.105, and the maximum deviation of ER_2_ is 121.0%. Therefore, changes in PLR have a significant impact on ER_2_.

When PNTD_2_ = 4.7 mm, the influence of changing PLR on ER_1_ and ER_2_ is shown in Figure 14. ER_1_ is almost unaffected by PLR, with the highest and lowest values of 1.092 and 1.090, respectively. In addition, when PLR increases from 102% to 104%, ER_2_ remains at 1.938; when PLR increases from 104% to 118%, ER_2_ decreases to the minimum value of 1.253, and the maximum deviation of ER_2_ is 54.7%. Therefore, the change of PLR has a relatively obvious impact on ER_2_. Moreover, it can be seen that compared with the ER value of PNTD_2_ = 4.1 mm, ER_1_ increases a lot, while ER_2_ decreases a little.

The effect of the changing PLR on ER_1_ and ER_2_ at PNTD_2_ = 5.3 mm is shown in Figure 15. With the increase of PLR, ER_1_ is still almost unaffected. However, compared with PNTD_2_ = 4.7 mm, ER_1_ increases largely. The maximum value of ER_2_ is lower than that of PNTD_2_ = 4.7 mm. When PLR increases from 102% to 106%, ER_2_ remains at 1.654 and it still shows a downward trend while PLR increases from 106% to 118%, and its maximum and minimum values are 1.654 and 1.213, respectively. The maximum deviation of ER_2_ is 36.4%. It is noted that at PNTD_2_ = 5.3 mm, the PLR affects ER_2_ to a smaller extent than at PNTD_2_ = 4.7 mm.

In summary, the change of PLR basically has no effect on the performance of the first stage and a significant effect on the performance of the second stage. Furthermore, the effect of PLR on the second-stage performance gradually diminishes with increasing PNTD_2_. It can also be seen that when PNTD_2_ = 4.1 mm, the ejector with a PLR of 102% is already in subcritical mode. When PNTD_2_ = 4.7 mm and PLR is 102–104%, the second-stage ejector is in critical mode, and when PLR is greater than 104%, it is in subcritical mode. When PNTD_2_ = 5.3 mm, the PLR is in the range of 102–106%, and the ejector is in critical mode. Therefore, the critical back pressure increases with the increase of PNTD_2_. The reason for this phenomenon is that, when the PNTD_2_ increases, which means the area ratio of the secondary stage reduces, normally the entrainment ratio increases with the area ratio when the back pressure keeps unchanged; as a result, the critical pressure increases with the increase of PNTD_2_. For different PNTD_2_, the next study will be carried out for the optimization of the key geometries of the second-stage ejector under different PLR, such as AR_2_ and NXP_2_, to identify the influence of back pressure and PNTD_2_ on the best AR_2_ and NXP_2_.

### 4.3. Effect of the Variable Back Pressure on the Key Geometries of the TSE

#### 4.3.1. Optimized AR_2_

The effect of AR_2_ at PNTD_2_ = 4.1 mm on ER_2_ at different PLR is shown in Figure 16. It can be seen that the change trend of ER_2_ affected by AR_2_ under the three PLR is consistent, that is, ER_2_ first increases, and then decreases as AR_2_ increases. At PLR = 102%, ER_2_ reaches its maximum value of 3.354 at AR_2_ = 23.5, so the best value for AR_2_ is 23.5, which is 13.0 times more than the optimal value of AR_2_ at PLR = 108%, and ER_2_ increases by 37.3% compared to the optimal value of 2.442 at PLR = 108%. When PLR is 104%, ER_2_ reaches a maximum value of 2.682 at AR_2_ = 17.5, then the optimal value of AR_2_ is 17.5, which is 7.0 times greater than the optimal value of AR_2_ at PLR = 108%; the optimal value of ER_2_ increases by 13.1% compared to the optimal value of 2.372 for ER_2_ at PLR = 108%. At PLR = 106%, the optimal value of 2.480 for ER_2_ is obtained at AR_2_ = 14.5, so that the optimal value for AR_2_ is 14.5, which is 4.0 times greater than the optimal value for AR_2_ at PLR = 108%, and the optimal ER_2_ increases by 9.4% over the optimal value for ER_2_ (2.266) at PLR = 108%. In summary, with the PLR increasing from 102% to 108%, the maximum ER_2_ decreases from 3.354 to 2.266, and the corresponding optimal AR_2_ decreases from 23.5 to 10.5.

Figure 17 shows the effect of AR_2_ at PNTD_2_ = 4.7 mm on ER_2_ under the three PLR. When PLR is 102%, ER_2_ increases first and then decreases with the change of AR_2_, and its maximum value of 3.022 is obtained at AR_2_ = 20.1, which is 55.9% higher than the optimal value of ER_2_ (1.938) at PLR of 108%; and the optimal AR_2_ is 12.0 times larger than that at PLR of 108%. With a PLR of 104%, when AR_2_ changes, the optimal value for AR_2_ is 16.1, and its corresponding maximum ER_2_ of 2.598, and ER_2_ increases by 0.662 over the optimal value of ER_2_ (1.936) at a PLR of 108%. When the PLR is 106%, and when AR_2_ changes, ER_2_ rises first and then decreases; its optimal value of 2.283 is obtained at AR_2_ = 13.1; and the optimal value of ER_2_ is increased by 0.368 compared to the optimal value of ER_2_ of 1.915 (PLR of 108%). Similar to PNTD_2_ = 4.1 mm, the maximum value of ER_2_ decreases and the corresponding optimal AR_2_ decreases as the PLR increases from 102% to 108%.

Figure 18 shows the change of ER_2_ at PNTD_2_ = 5.3 mm with AR_2_ under the different PLR conditions. At PLR = 102%, as AR_2_ changes from 2.9 to 19.9, ER_2_ increases from 0.336 to a maximum of 2.663 at AR_2_ = 16.9, thus, the best value for AR_2_ is 16.9. Furthermore, the optimal value of ER_2_ is increased by 1.008 when compared to the optimal ER_2_ (1.655) at PLR = 108%. At PLR = 104%, ER_2_ achieves a maximum value of 2.299 at AR_2_ = 12.9. Therefore, the optimal AR_2_ is 12.9. In addition, the optimal value of ER_2_ is increased by 0.645 when compared to the best ER_2_ (1.654) at PLR = 108%. When PLR = 106%, ER_2_ shows a trend of increasing first and then decreasing with the increase of AR_2_. Its maximum value 2.050 is obtained at AR_2_ = 10.9, hence, the optimum of AR_2_ is 10.9. Compared with the optimal ER_2_ (1.653) when PLR is 108%, the maximum ER_2_ increases by 0.397. It can be seen that when PNTD_2_ is 5.3 mm and PLR changes from 102% to 108%, the maximum value of ER_2_ and the corresponding optimal AR_2_ also show a decreasing trend. This phenomenon can be probably explained as follows: when the PLR increases, the pressure difference between the outlet and the secondary flow inlet pressure increases, in order to maintain the increased pressure lift; thus, the ER_2_ usually drops and the suitable AR_2_ reduces accordingly.

#### 4.3.2. Optimized NXP_2_

Figure 19 shows the influence of NXP_2_ on ER_2_ under different PLR when PNTD_2_ is 4.1 mm. With given PLR = 102%, as NXP_2_ increases from 18 mm to 30 mm, the change trend of ER_2_ rises first and then decreases. Its maximum value of 2.486 is obtained when NXP_2_ = 26 mm, and the maximum deviation of ER_2_ is 1.944%. With given PLR = 104%, as NXP_2_ increases from 18 mm to 26 mm, ER_2_ increases from 2.341 to 2.382 and then decreases to 2.369, so the maximum deviation of ER_2_ is 1.756%. With given PLR = 106%, ER_2_ also shows a trend of first increasing and then decreasing, and its peak value of 2.269 is obtained at NXP_2_ = 26 mm, and the maximum deviation of ER_2_ is 1.584%. Therefore, it can be seen that when PNTD_2_ = 4.1 mm, the optimal value of NXP_2_ does not change with the change of PLR, and thus the influence of NXP_2_ on ER_2_ is far less than that of AR_2_.

Figure 20 demonstrates the influence of NXP_2_ on ER_2_ under different PLR when PNTD_2_ is 4.7 mm. When PLR is 102%, as NXP_2_ changes from 18 mm to 24 mm, ER_2_ rises from 1.932 to 1.950, and then decreases to 1.929 when NXP_2_ is 30 mm, and the maximum deviation of ER_2_ is 1.057%. When PLR is 104%, the change trend of ER_2_ also increases first and then decreases, and its peak value of 1.936 appears at the value of 24 mm of NXP_2_, while the maximum deviation of ER_2_ is 0.868%. When PLR is 106%, the variation trend of ER_2_ is similar to the previous two, and its maximum value of 1.921 is obtained at NXP_2_ = 24 mm, and the maximum deviation of ER_2_ is 0.744%. Therefore, when PNTD_2_ is 4.7 mm, the optimal value of NXP_2_ is 24 mm, which does not change with the change of PLR.

Figure 21 shows the influence of NXP_2_ on ER_2_ under different PLR when PNTD_2_ is 5.3 mm. The changing trend of ER_2_ is the same under the three PLR conditions, that is, ER_2_ increases first and then decreases, and all the maximum value of ER_2_ appear at NXP_2_ = 22 mm. Therefore, when PNTD_2_ is 5.3 mm, the best value of NXP_2_ is 22 mm, that is, it is not affected by the change of PLR.

## 5. Conclusions

In this paper, the CFD simulation method was first used to optimize the nozzle geometry of the second-stage ejector under different PNTD_2_ with the operating conditions given in Table 1. Then, the effect of variable back pressure on the ejector performance was studied. Finally, three PLRs that place the ejector in critical or near critical mode were selected to study the influence of AR_2_ and NXP_2_ on the performance of the second-stage ejector. The main findings obtained are as follows:(1)When LC_2_ and AC_2_ change, the maximum values of ER_1_ and ER_2_ do not appear at the same length or angle. LC_2_ = 25 mm and AC_2_ = 12° are the relative optimal combination of values for the second-stage ejector nozzle;(2)With the increase of PNTD_2_, ER_1_ and ER_2_ decrease with the increase of LC_2_ and AC_2_, and the PLR range of the ejector in the critical mode gradually increases;(3)The change of PLR has no effect on the performance of the first-stage ejector, but has a significant effect on the performance of the second-stage ejector; with the increase of PNTD_2_, the influence of PLR on the performance of the second stage is gradually weakened;(4)When PNTD_2_ is 4.1 mm, the optimal value of AR_2_ decreases from 23.5 to 14.5 with the increase of PLR, and the peak value of ER_2_ decreases from 3.354 to 2.480. When PNTD_2_ is 4.7 mm, the optimal value of AR_2_ decreases from 20.1 to 13.1 with the increase of PLR, and the maximum value of ER_2_ decreases from 3.022 to 2.383. When PNTD_2_ is 5.3 mm, with the change of PLR from 102% to 106%, the optimal value of AR_2_ is from 16.9 to 10.9, and the peak value of ER_2_ is reduced from 3.354 to 2.382;(5)The optimal value of NXP_2_ is not affected by the change of PLR. When PNTD_2_ is 4.1 mm, 4.7 mm and 5.3 mm, the corresponding optimal value of NXP_2_ is 26 mm, 24 mm and 22 mm, respectively.

## Figures and Tables

**Figure 1 entropy-24-01847-f001:**
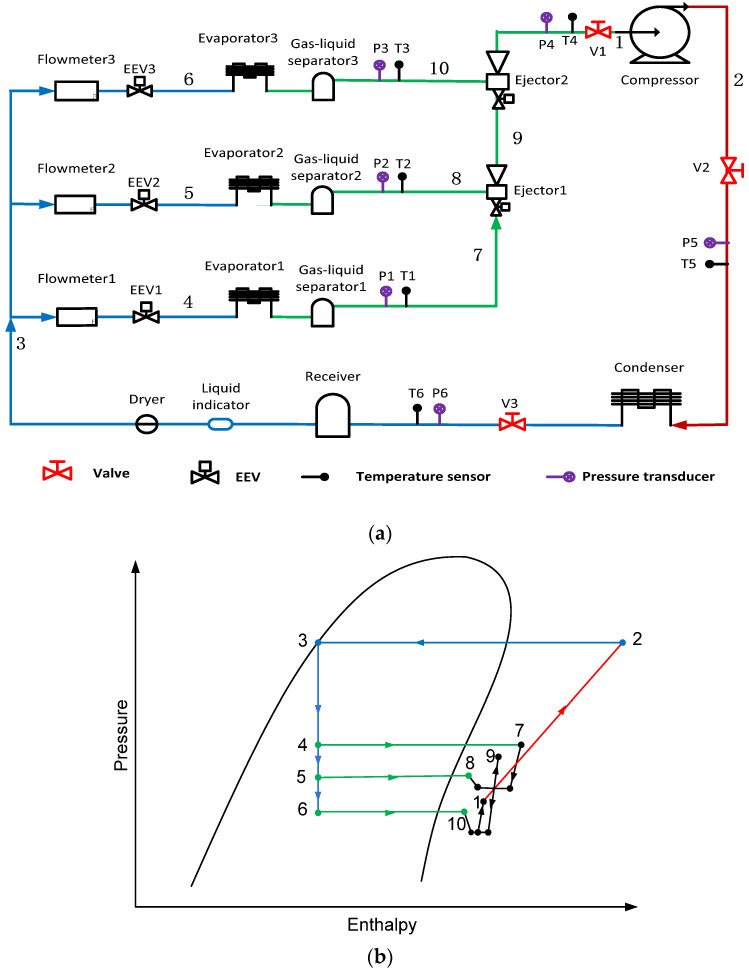
The schematic of the highly coupled TSE-based refrigeration system: (**a**) schematic; and (**b**) pressure-enthalpy diagram.

**Figure 2 entropy-24-01847-f002:**
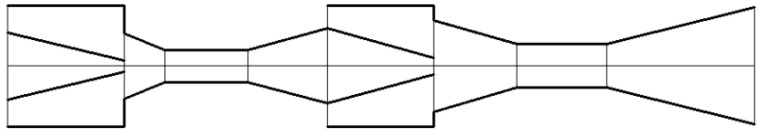
The schematic of the highly coupled TSE.

**Figure 3 entropy-24-01847-f003:**
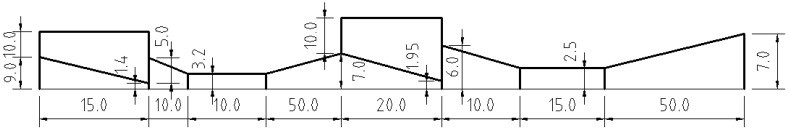
Initial geometrical parameters of the TSE.

**Figure 4 entropy-24-01847-f004:**
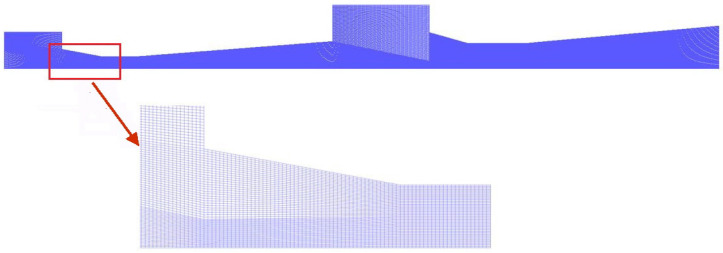
2D axisymmetric quadrilateral grids of the TSE.

**Figure 5 entropy-24-01847-f005:**
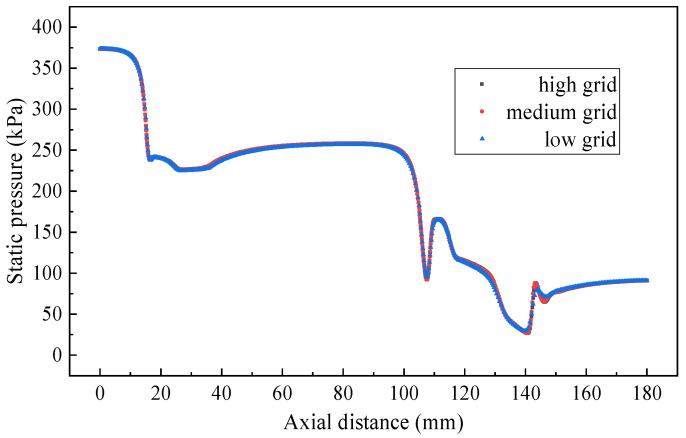
Axial static pressure with grid levels of the TSE.

**Figure 6 entropy-24-01847-f006:**
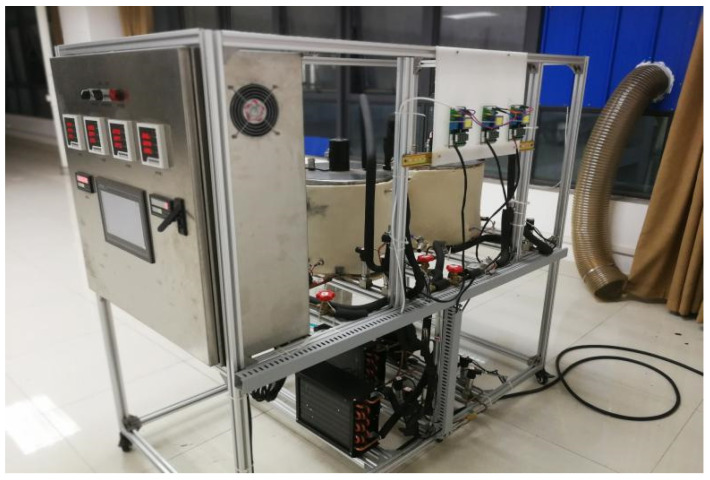
Photograph of experimental setup.

**Figure 7 entropy-24-01847-f007:**
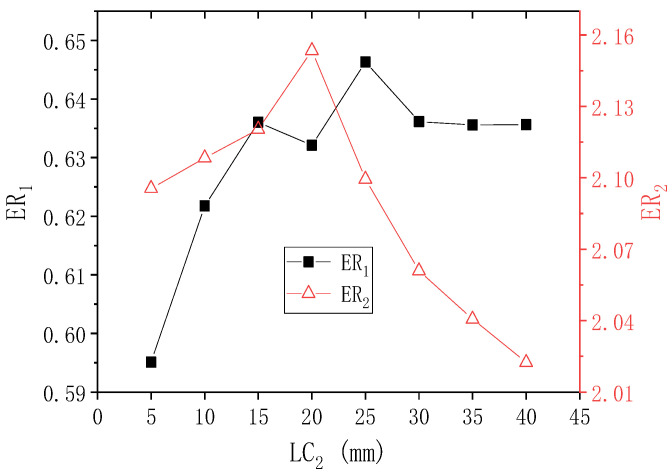
Changes of ER_1_ and ER_2_ with LC_2_ when PNTD_2_ is 4.1 mm.

**Figure 8 entropy-24-01847-f008:**
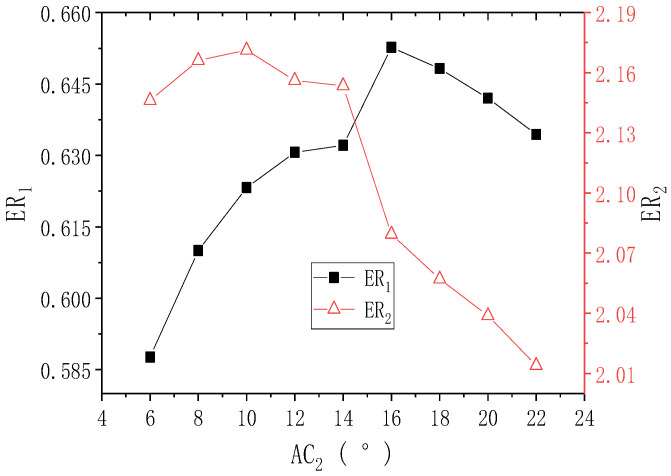
Changes of ER_1_ and ER_2_ with AC_2_ when PNTD_2_ is 4.1 mm.

**Figure 9 entropy-24-01847-f009:**
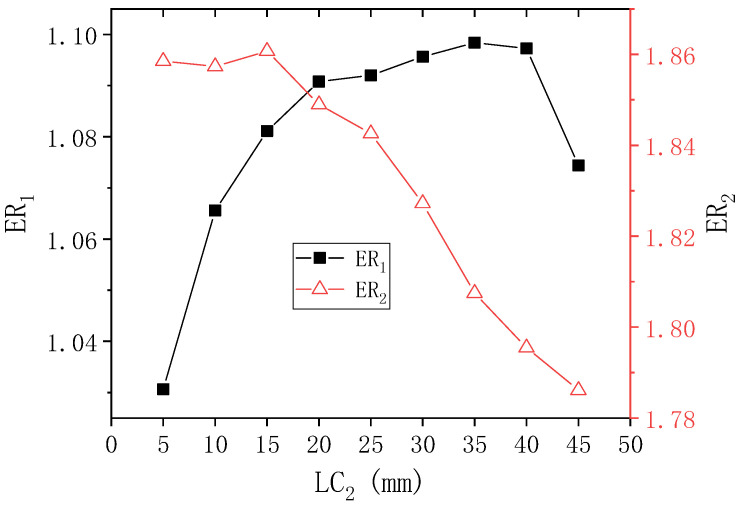
Changes of ER_1_ and ER_2_ with LC_2_ when PNTD_2_ is 4.7 mm.

**Figure 10 entropy-24-01847-f010:**
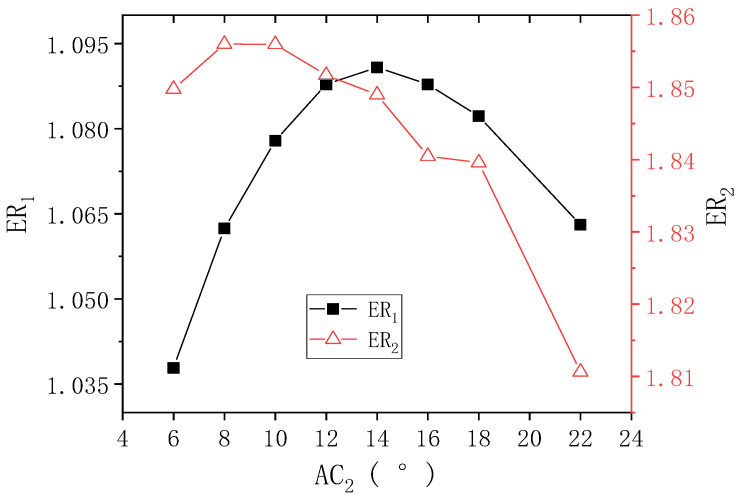
Changes of ER_1_ and ER_2_ with AC_2_ when PNTD_2_ is 4.7 mm.

**Figure 11 entropy-24-01847-f011:**
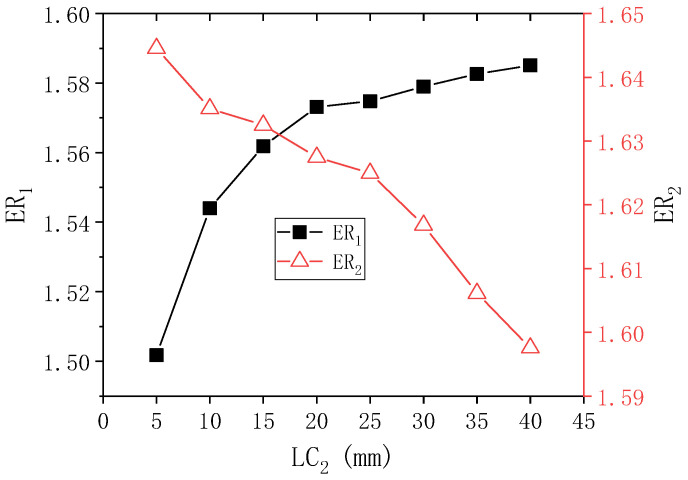
Changes of ER_1_ and ER_2_ with LC_2_ when PNTD_2_ is 5.3 mm.

**Figure 12 entropy-24-01847-f012:**
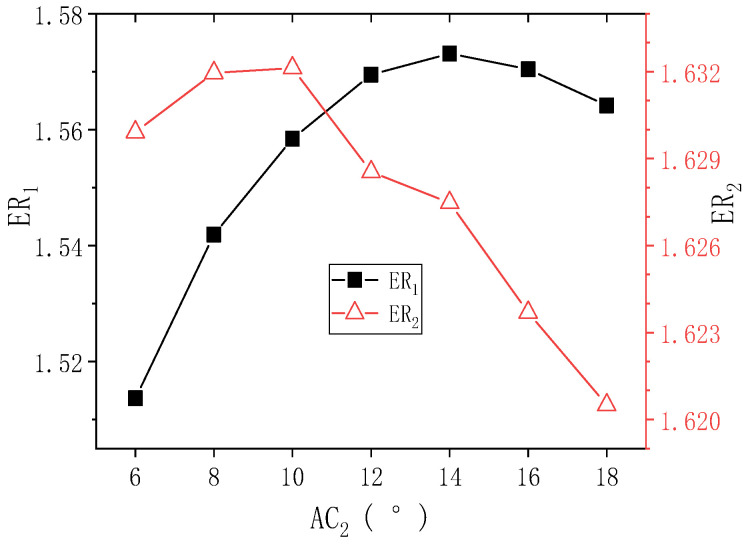
Changes of ER_1_ and ER_2_ with AC_2_ when PNTD_2_ is 5.3 mm.

**Figure 13 entropy-24-01847-f013:**
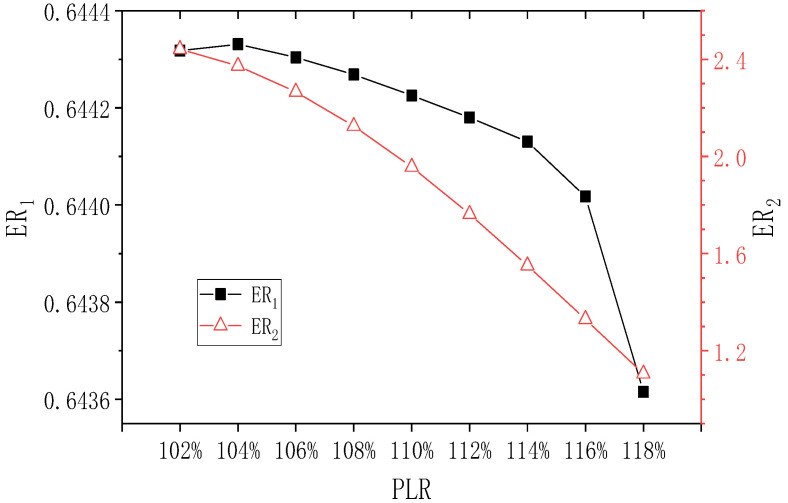
Effect of varied PLR on ER_1_ and ER_2_ (PNTD_2_ = 4.1 mm).

**Figure 14 entropy-24-01847-f014:**
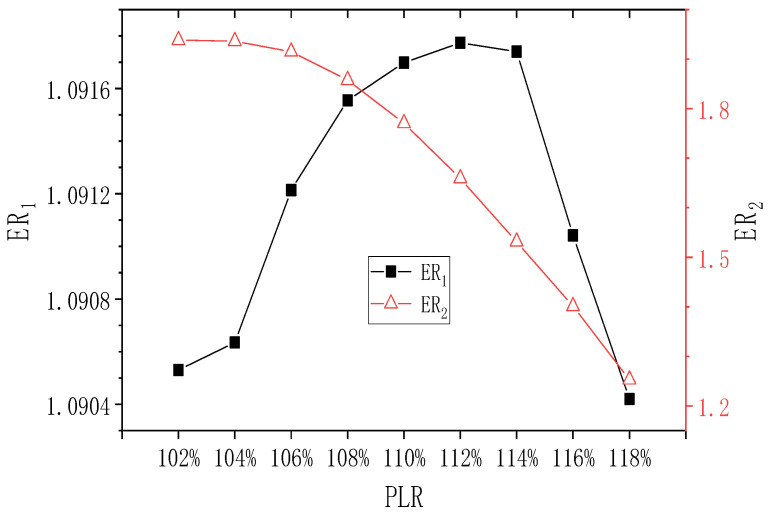
Effect of varied PLR on ER_1_ and ER_2_ (PNTD_2_ = 4.7 mm).

**Figure 15 entropy-24-01847-f015:**
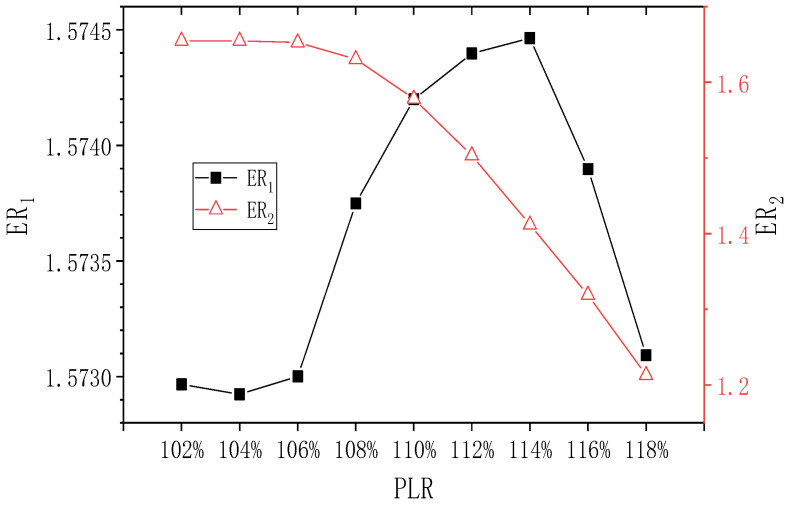
Effect of varied PLR on ER_1_ and ER_2_ (PNTD_2_ = 5.3 mm).

**Figure 16 entropy-24-01847-f016:**
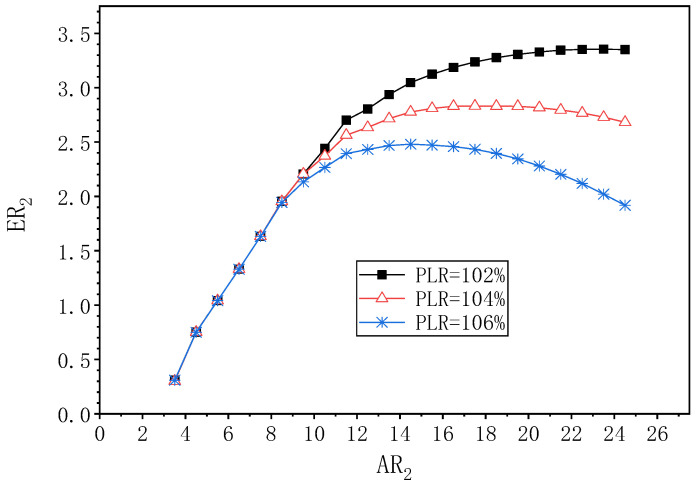
The variation of ER_2_ with AR_2_ under different PLR (PNTD_2_ = 4.1 mm).

**Figure 17 entropy-24-01847-f017:**
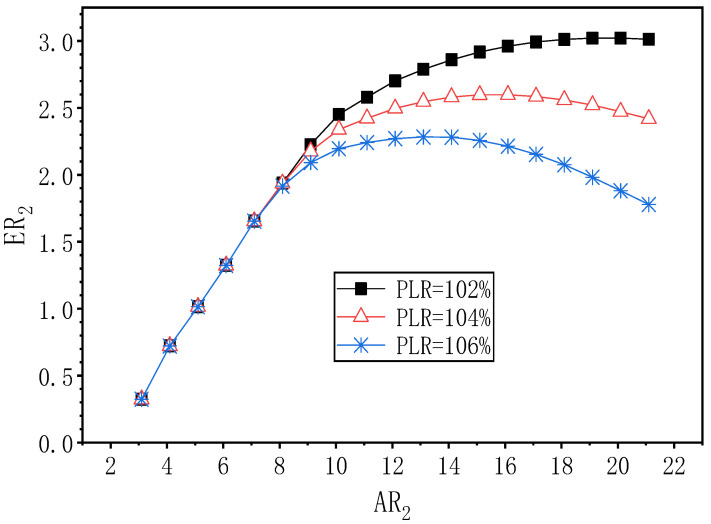
The variation of ER_2_ with AR_2_ under different PLR (PNTD_2_ = 4.7 mm).

**Figure 18 entropy-24-01847-f018:**
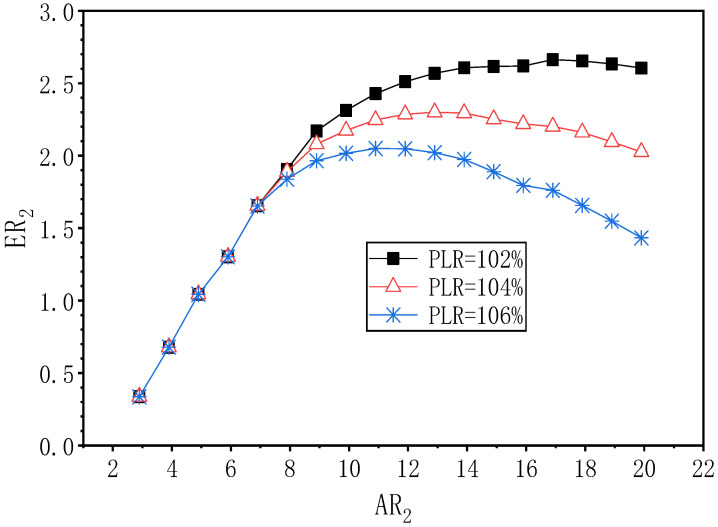
The variation of ER_2_ with AR_2_ under different PLR (PNTD_2_ = 5.3 mm).

**Figure 19 entropy-24-01847-f019:**
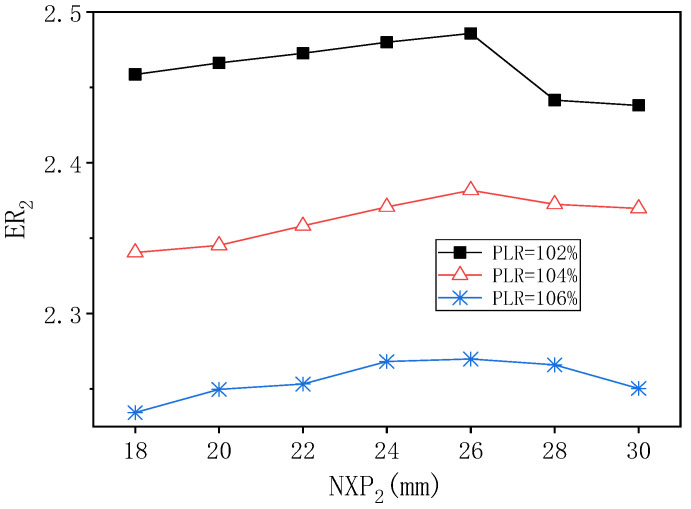
The variation of ER_2_ with NXP_2_ under different PLR (PNTD_2_ = 4.1 mm).

**Figure 20 entropy-24-01847-f020:**
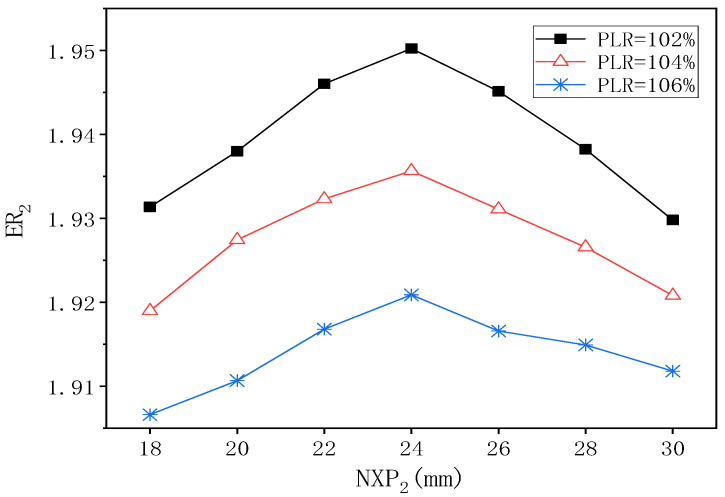
The variation of ER_2_ with NXP_2_ under different PLR (PNTD_2_ = 4.7 mm).

**Figure 21 entropy-24-01847-f021:**
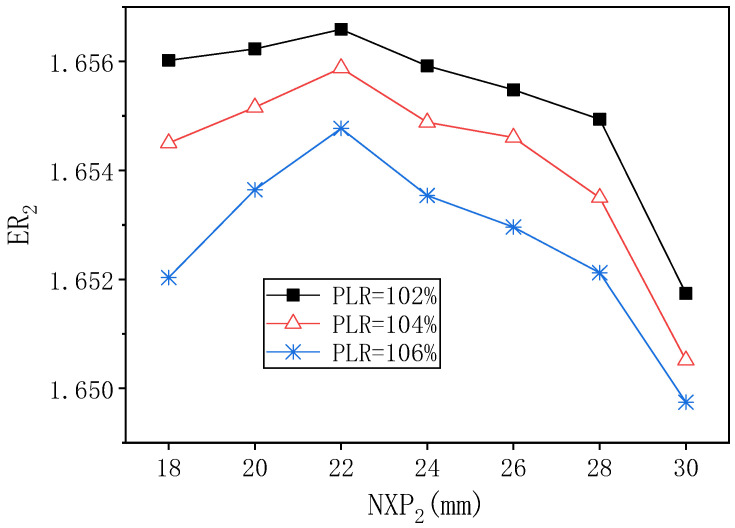
The variation of ER_2_ with NXP_2_ under different PLR (PNTD_2_ = 5.3 mm).

**Table 1 entropy-24-01847-t001:** Boundary conditions of the TSE.

Boundary Conditions	P (kPa)	T (K)
First-stage primary flow	374.6	290
First-stage secondary flow	243.4	278
Second-stage secondary flow	84.4	253
Outlet flow	91.15	260

**Table 2 entropy-24-01847-t002:** Position, range and accuracy of sensors.

Sensors	Position	Unit	Range	Accuracy
Temperature	T_1_, T_2_, T_3_	°C	−40~40	±0.3
	T_4_, T_5_, T_6_	°C	0~100	±0.3
Pressure	P_1_, P_2_, P_3_	Bar	−1~8	±0.5%
	P_4_, P_5_, P_6_	Bar	−1~16	±0.5%
Volume flow rate	Flowmeter 1	L/h	6~60	±1.6%
	Flowmeter 2	L/h	6~60	±1.6%
	Flowmeter 3	L/h	6~60	±1.6%

**Table 3 entropy-24-01847-t003:** Operating conditions of the TSE for CFD model validation.

	First-Stage Primary Flow		First-Stage Secondary Flow		Second-Stage Secondary Flow		Outflow
	P	T	P	T	P	T	P
	(kPa)	(K)	(kPa)	(K)	(kPa)	(K)	(kPa)
Group 1	414.6	294	243.4	278	84.4	253	91.15
	394.6	292					
	374.6	290					
	354.6	288					
	334.6	286					
Group 2	374.6	290	283.4	282	84.4	253	
			263.4	280			
			243.4	278			
			223.4	276			
			203.4	274			
Group 3	374.6	290	243.4	278	88.4	254	
					86.4	253.5	
					84.4	253	
					82.4	252.5	
					80.4	252	

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
