# Peer review of "Effect of Back Pressure on Performances and Key Geometries of the Second Stage in a Highly Coupled Two-Stage Ejector"

_entropy, 2022, doi:10.3390/e24121847_

Round 1

Reviewer 1 Report

Here follow my main remarks on this paper:

1) The parts related to the description and validation of the model are quite scarce and would benefit from more details. The sketch is clear, but a thermodynamic diagram showing the processes would be of use. 

2) Also sections 2 and 3 are extremely poor, even if they quote preceding articles.

3) Some typos are present. Please check carefully all the main text

4) The first sentence of the abstract does not make sense "In this paper,..[]...conducted first". It should be rephrased

5) The state of the art might be improved by also indicating the main results of each paper, and also by quoted recent works published on MDPI and dealing with the modeling of ejector cycles (see for instance https://doi.org/10.3390/en14185663 https://doi.org/10.3390/en13030562)

6) A nomenclature is needed, and some acronyms are not explained. Moreover, there is no definition for the reference parameters (for instance Entrainment ratio ER) related to each ejector phase. How the total pressure lift is calculated?

7) Figure 5 obtained from the data of Table 1 is the same of a preceding article by the same authors.

8) In the results section, the operating conditions used for the simulations should be indicated.

9) some results are not trivial, and the authors should attempt a physical explanation. It seems that the comments are only a detailed description of trends.

Reviewer 2 Report

In this paper, the authors proposed a highly coupled two-stage ejector-based cooling cycle, and carried out the optimization of primary nozzle geometry of the second-stage ejector under varied primary nozzle of the second-stage, in addition the identification of the effect of the variable back pressure on the key geometries of the two-stage ejector was carried out. The works of this paper is worth to be investigated and is meaningful. However, the following comments should be addressed by the authors:

1. Nomenclature needs to be given in this paper.

2. The detailed connection between the first-stage and the second-stage in the Figure 2 needs to be stated.

3. More detailed description of the CFD simulation settings should be presented in the Section 2.2.

4. In Figure 7, please analyze why the ER1 of the point of LC2=20mm is abnormal as compared to other points.

5. Please give out the analysis for the relations between Figures 7-9.

6. Please present more detailed analysis for the differences between Figures 13-15.

7. Please improve the language of the English.

8. Some reference about condensation in ejector can be discussed

Round 2

Reviewer 1 Report

No more suggestions. Congratulations for your work

Reviewer 2 Report

accept